# Detection of Apparent Early Rabies Infection by LN34 Pan-Lyssavirus Real-Time RT-PCR Assay in Pennsylvania

**DOI:** 10.3390/v14091845

**Published:** 2022-08-23

**Authors:** Lisa Dettinger, Crystal M. Gigante, Maria Sellard, Melanie Seiders, Puja Patel, Lillian A. Orciari, Pamela Yager, James Lute, Annette Regec, Yu Li, Dongxiang Xia

**Affiliations:** 1Bureau of Laboratories, Pennsylvania Department of Health, Exton, PA 19341, USA; 2Division of High-Consequence Pathogens and Pathology, Centers for Disease Control and Prevention, Atlanta, GA 30329, USA

**Keywords:** rabies virus, reverse transcriptase polymerase chain reaction (RT-PCR), real-time polymerase chain reaction (qPCR or rtPCR), lyssavirus, rabies diagnostics, DFA, pan-lyssavirus, LN34 rRT-PCR, early rabies infection, RNA

## Abstract

The Pennsylvania Department of Health Bureau of Laboratories (PABOL) tested 6855 animal samples for rabies using both the direct fluorescent antibody test (DFA) and LN34 pan-lyssavirus reverse transcriptase quantitative PCR (RT-qPCR) during 2017–2019. Only two samples (0.03%) were initially DFA negative but positive by LN34 RT-qPCR. Both cases were confirmed positive upon re-testing at PABOL and confirmatory testing at the Centers for Disease Control and Prevention by LN34 RT-qPCR and DFA. Rabies virus sequences from one sample were distinct from all positive samples processed at PABOL within two weeks, ruling out cross-contamination. Levels of rabies virus antigen and RNA were low in all brain structures tested, but were higher in brain stem and rostral spinal cord than in cerebellum, hippocampus or cortex. Taken together, the low level of rabies virus combined with higher abundance in more caudal brain structures suggest early infection. These cases highlight the increased sensitivity and ease of interpretation of LN34 RT-qPCR for low positive cases.

## 1. Introduction

Rabies is a fatal but preventable infectious disease that causes approximately 60,000 human deaths worldwide each year [1]. In the United States, rabies causes few human deaths due to the elimination of rabies variants maintained in domestic dogs and large-scale, sustained rabies control efforts [2,3,4]. However, rabies endemic in wildlife still presents a threat to humans and domestic animals. Rabies surveillance in the United States involves over 125 testing laboratories that test approximately 100,000 animals and identify 5000 rabid animals each year [5,6]. 

In the United States, rabies diagnostic testing is predominantly performed using the Direct Fluorescent Antibody test (DFA). DFA has been a reliable and sensitive rabies diagnostic test for over 60 years. The World Health Organization (WHO) and World Organization for Animal Health (OIE) recognize the DFA, the direct rapid immunohistochemical test, and reverse transcriptase polymerase chain reaction (RT-PCR) as acceptable diagnostic tests for the detection of rabies virus [7,8]; however, the DFA is the only test currently recommended for primary rabies diagnosis in the United States [9]. Molecular methods such as reverse transcriptase quantitative PCR (RT-qPCR) provide several advantages over DFA testing. Many public health laboratories routinely perform RT-qPCR for detection of other pathogens and already have the equipment and expertise to implement a rabies RT-qPCR test. DFA, however, requires fluorescence microscopy expertise, which is less frequently used in diagnosis of other pathogens. 

RT-qPCR is not currently recommended for primary diagnostic testing of rabies samples in the United States, though it can be used as a confirmatory test [9]. Many laboratories across the country are currently implementing the LN34 RT-qPCR assay [10,11] for confirmatory rabies testing. The Pennsylvania Department of Health Bureau of Laboratories (PABOL) routinely tests every rabies suspect sample by both DFA and LN34 RT-qPCR. PABOL tests approximately 2800 animals associated with human exposures for rabies annually. On average, 114 positive rabies cases are identified (4%). Raccoons and bats are the major reservoirs in Pennsylvania, and Eastern Raccoon and several bat rabies virus variants are endemic. 

During 2017–2019, 6855 animals were tested by both DFA and PCR at PABOL. Of those tested, only two (0.03%) were initially DFA negative but positive by LN34 RT-qPCR. In June 2017, an adult raccoon in Carbon County, PA, displaying clinical manifestation of rabies, attacked a chicken and charged an individual. On 1 June, the animal was euthanized and submitted for rabies testing. Initial DFA testing produced a negative result, but LN34 RT-qPCR was positive. In June 2019, a mother raccoon was hit and killed on a road in Venango County, PA, leaving behind two young offspring. The two juvenile raccoons were taken into a home and kept from 11 June to 12 June, during which time they were handled by four persons. On 13 June, both juveniles were euthanized and submitted for rabies testing. One juvenile was negative by both DFA and LN34 RT-qPCR. The other juvenile tested positive by LN34 RT-qPCR after initial negative DFA result. The following report describes the subsequent investigation into the two discordant cases.

## 2. Materials and Methods

### 2.1. Samples

Samples were submitted to PABOL as part of routine rabies surveillance and diagnostic testing. Animal collection was not performed as part of this study; therefore, institutional animal care and use committee approval was not necessary. 

### 2.2. Direct Fluorescent Antibody (DFA) Test

PABOL: Brain tissue representing a full transverse cross section of brain stem and three lobes of cerebellum and/or hippocampi were minced together. These brain tissue preparations were tested using a modification of the Protocol for Postmortem Diagnosis of Rabies in Animals by Direct Fluorescent Antibody Testing, a minimum standard for rabies diagnosis in the United States (national standard protocol) [9]. Additional details can be found in Appendix A.

CDC: Samples were tested according to the national standard protocol and Direct Fluorescent Antibody Test, WHO, Laboratory Techniques in Rabies [9,12]. Additional details can be found in Appendix A.

### 2.3. Real-Time RT-PCR (RT-qPCR)

Tissue representing a full cross section of the brain stem and all three lobes of the cerebellum was transferred to TRIzol Reagent (Life Technologies 15596018) and then extracted using Direct-zol RNA MiniPrep kit (R2052 Zymo, Irvine, CA, USA) following the published protocol for LN34 RT-qPCR [11]. Additional or repeat RT-qPCR testing of separate tissues was performed for the brain stem, rostral spinal cord, cerebellum, hippocampi and cortex. Samples were tested in duplicate on the Applied Biosystems 7500 Fast Dx platform at PABOL. Samples were tested in triplicate on Applied Biosystems ViiA7 platform at CDC. LN34 Cq values were used to compare relative levels of viral RNA in different brain regions. Seven salivary gland (left and right sides), three oral swabs, one nasal swab and three muzzle skin (left and right sides) samples were collected from the head of the 2017 raccoon; samples were extracted and tested by LN34 RT-qPCR as described above for rabies at CDC. CDC operators were not blinded to the PABOL results.

Quantification of RT-qPCR results was performed using the delta delta Cq method (ΔΔCq or ddCq) [13]. Average LN34 and beta actin Cq values were calculated for each brain region examined. Average beta actin Cq value was subtracted from the average LN34 Cq value for each brain region to calculate the ΔCq. The brain stem was chosen as the reference tissue, so ΔCq for the brain stem was subtracted from ΔCq for other brain regions to calculate ΔΔCq for each brain region. The amount of target was estimated as 1.93-ΔΔCq based on the efficiency of the LN34 assay as 93% for rabies virus based on previous estimation [11]. The plot was generated in RStudio [14] using ggplot2 [15] and finished in Inkscape 0.91 (inkscape.org, accessed on 6 August 2019).

### 2.4. Sequencing

Rabies virus sequencing was performed at CDC for the 2019 juvenile raccoon case and four additional positive samples that were processed at PABOL within two weeks of the 2019 case. Rabies virus RNA was extracted using Direct-zol RNA MiniPrep kit (R2052 Zymo, Irvine, CA, USA), and complete nucleoprotein and glycoprotein genes were amplified using Takara long amplicon Taq polymerase with GC buffers (RR02AG Takara Bio USA, Mountain View, CA, USA) using the primers indicated in Table 1 after cDNA synthesis using random hexamer primers and Roche AMV reverse transcriptase (10109118001 Roche, Sigma-Aldrich, St. Louis, MO, USA). Samples were multiplexed using Takara long amplicon Taq polymerase with GC buffers following the manufacturer’s instructions for PCR barcoding for nanopore sequencing (EXP-PBC096 Oxford Nanopore Technologies, Oxford, UK). Samples were pooled and sequenced using the Oxford Nanopore MinION, following the manufacturer’s instructions for the ligation sequencing kit (SQK-LSK108 Oxford Nanopore Technologies, Oxford, UK). Consensus sequences were generated in CLC Genomics Workbench 12 (Qiagen, Venlo, The Netherlands) after read mapping to rabies virus reference genomes using bwa mem -x ont2d and were polished using nanopolish version 0.6.0 (https://github.com/jts/nanopolish/, accessed on 23 March 2020). Manual indel correction was then performed as described previously for the coding regions of the nucleoprotein and glycoprotein genes [16]. Sequence differences were determined based on coding region alignments generated using mafft v7.308 [17,18] in geneious 9.1.4 (Biomatters, Inc., Newark, NJ, USA). Phylogenetic analysis was performed by Maximum Likelihood in Mega 7.0.26 [19] using GTR + G + I model of evolution, which was determined using model test in Mega7. Sequences were deposited to GenBank under accession numbers OP221406 - OP221415. 

## 3. Results

### 3.1. PABOL DFA and PCR Testing

During 2017–2019, PABOL tested 6855 animal samples submitted for rabies testing. A total of 342 samples were positive (4.06%). Raccoon was identified as the leading host species, with 123 rabid raccoons identified, followed by cats (91), foxes (45), bats (43) and skunks (17) (Figure 1). 

Since 2018, PABOL has routinely tested all rabies samples by DFA in parallel with LN34 RT-qPCR. PABOL participated in an LN34 RT-qPCR pilot study with the Centers for Disease Control and Prevention (CDC) in 2016 [11] and fully implemented PCR along with DFA testing for all samples in 2018. Among 6855 samples tested in 2017–2019, discordant results were identified for only two cases (0.03%): an adult raccoon tested in 2017 (sample 1130) and a juvenile raccoon tested in 2019 (sample 1059). In these two cases, the initial DFA tests were negative for rabies antigen; however, rabies virus RNA was detected by LN34 RT-qPCR (Table 2).

In both cases, the original tissues were reprocessed, taking separate samples from different regions of the brain, including the rostral spinal cord, brain stem, cerebellum, and hippocampus. These separate brain tissues were tested by both DFA and LN34 RT-qPCR. Upon re-testing, some atypical, sparse staining was observed by DFA in the brain stem and spinal cord impressions but was notably absent from the cerebellum and hippocampus. Rabies virus RNA levels were low in all tissues tested for RT-qPCR, with the highest levels (lowest quantification cycle (Cq) values) in the spinal cord and brain stem and the lowest levels in the cerebellum and hippocampus (Table 2).

### 3.2. CDC DFA and PCR Testing

Brain samples were sent to the Poxvirus and Rabies Branch at CDC for confirmatory testing by DFA and LN34 RT-qPCR. Tissues from both cases were confirmed positive with low antigen distribution; however, antigen distribution varied in different regions of the brain (Table 3). For both cases, all impressions prepared from the brain stem or rostral spinal cord tissue were positive, with typical antigen in <10% of fields examined. Cerebellum tissue also produced positive DFA results; however, a typical rabies antigen was observed in only 2/6 slides for the 2017 adult raccoon and 3/5 slides for the 2019 juvenile raccoon. Rabies antigen distribution in the positive cerebellum slides was also in <10% of fields. Impressions from cortex and hippocampus were tested from the 2019 juvenile raccoon. One slide out of six showed atypical staining; the remaining five cortex/hippocampus slides did not contain typical rabies antigens, and the result was indeterminate. All samples exhibited amplification by LN34 RT-qPCR, indicating the presence of rabies virus RNA. In some cases, amplification did not reach the threshold or the Cq value was later than the cut-off for a positive sample (Cq ≤ 35 [11]), indicating an indeterminate result (Table 3). Rabies virus RNA levels were highest in the spinal cord and brain stem and lowest in the cortex/hippocampus (Figure 2).

Additional RT-qPCR testing was performed on seven salivary gland, three oral swabs, one nasal swab and three muzzle skin samples from the 2017 raccoon. No rabies virus RNA was detected in oral, nasal, salivary gland or skin samples from the 2017 raccoon (42 replicates for 14 samples).

### 3.3. Investigation into Potential Cross-Contamination

Taken together, the low overall rabies virus RNA level and distribution pattern (highest levels in caudal brain regions and lowest in rostral regions) in these two cases could indicate early infection or cross-contamination. To rule-out the possibility of cross-contamination, rabies virus sequencing was performed on the 2019 juvenile case and all positive samples processed at PABOL within two weeks. These included grey fox sample 997 (processed 6/11), bat sample 1018 (processed 6/13), grey fox sample 1090 (processed 6/19), and cat sample 846 (used as a positive control the week juvenile raccoon 1059 was tested). Sequencing was not performed for the 2017 case because samples were no longer available.

Complete nucleoprotein and glycoprotein gene sequences were generated and compared to publicly available reference sequences from representative rabies virus variants. BLAST search of rabies virus sequences from the 2019 juvenile raccoon revealed > 99% nucleotide identity with Eastern Raccoon rabies virus variant isolates from the eastern US. Phylogenetic analysis revealed the 2019 juvenile raccoon sequence clustered with other Eastern Raccoon variant sequences from PA and reference sequence MK540681 (raccoon from NY 1991) (Figure 3). PA cat 846, PA fox 997 and PA fox 1090 clustered with the 2019 juvenile raccoon sequence within with the Eastern Raccoon rabies virus variant clade. PA bat 1018 clustered with reference JQ685920, collected from a big brown bat in PA in 1984 and rabies virus variant EF-E1 [11] that is maintained in the big brown bat, *Eptesicus fuscus*, in the eastern US.

The 2019 juvenile raccoon sequences exhibited many differences from all other PABOL samples processed within two weeks (Figure 3, Appendix A). The nucleoprotein gene had 17–23 nucleotide differences relative to the Eastern Raccoon variant samples and 195 differences relative to bat sample 1018. The glycoprotein gene had 23–26 changes from the Eastern Raccoon variant samples and 270 changes from the bat sample. The closest PABOL sequence was fox sample 997, which exhibited 98.7% and 98.5% identity to the nucleoprotein and glycoprotein genes, respectively. The 2019 juvenile raccoon sequences were more similar to an Eastern Raccoon variant isolate from NY in 1991 (MK540681, with 99.04% and 99.49% identity to nucleoprotein and glycoprotein genes, respectively). Taken together, these data suggest contamination was unlikely to be the cause of the positive PCR result.

## 4. Discussion

We describe two cases where LN34 RT-qPCR identified rabies infection with low viral RNA after initial DFA testing failed to detect the presence of rabies virus antigen. Repeat testing at PABOL and confirmatory testing at CDC confirmed both as positive rabies cases, and appropriate public health response was initiated. These cases highlight the sensitivity and objectivity of PCR in cases with low rabies virus antigen and RNA and, thus, support the use of RT-qPCR in routine rabies diagnostic testing.

A false negative result for a rabies diagnostic test is extremely serious because rabies is nearly always fatal if post exposure treatment is not administered promptly. The DFA has been used for over 60 years in the United States with no known deaths caused by failures to detect rabies cases. With these two cases, RT-qPCR demonstrated higher sensitivity than DFA at PABOL, and the reasons behind this are worth considering. 

PABOL tests thousands of samples each year, and the concordance rate for DFA with PCR was 99.97% for 6855 samples. If there was a systemic issue with DFA testing at PABOL, a lower concordance rate with LN34 would be expected, similar to what has been reported previously for laboratories with systemic DFA issues [11]. One observation worth noting is the practice of making impressions from minced brain tissues at PABOL. The United States national standard protocol [9] and WHO [12] recommend that impressions are taken directly from tissue for DFA testing. However, repeat testing of tissue impressions from these cases also produced negative DFA results at PABOL. During initial testing at PABOL by DFA, one slide containing the brain stem and cerebellum was tested for each sample and the results were negative. Prompted by the positive PCR result, DFA re-testing was initiated. Many impressions were made from different brain regions, increasing the opportunity to detect sparse antigens at both labs. However, the antigen was detected in every brain stem impression tested at CDC for both samples.

A United States national standard protocol [9] was developed to avoid differences between laboratories, and any differences in DFA test procedures between laboratories can affect test results [20,21] and should be avoided. The DFA procedure could vary between laboratories due to differences in commercial monoclonal antibody reagents or if optimal working dilutions of conjugate were not prepared properly [9,22]. Differences in fluorescence microscopes and objective lens quality could, in theory, produce different results for a sample with an extremely low antigen level. The DFA relies heavily on the expertise of the person interpreting results, who must be able to distinguish a typical fluorescent rabies virus antigen from non-specific fluorescent objects such as bacteria or artifacts in the tissue. All atypical, weak, or unusual tests are repeated using a specificity control or sent to CDC for confirmation. 

In contrast to DFA, PCR methods are easier to standardize, and result interpretation is inherently more objective. Primer and probe sequences and concentrations can be defined in protocols for high reproducibility between laboratories and uniformity between manufacturers and lots. Currently, CDC provides a standardized positive control to ensure proper performance of the LN34 across laboratories. Test output is a quantitative Cq value, which determines positive, negative, or indeterminate result based on its numeric value. 

The high sensitivity of PCR can lead to false positive results caused by cross-contamination, especially in laboratories inexperienced with PCR. In most cases, cross-contamination can be avoided through good laboratory practices. Cross-contamination can occur during tissue processing, sample preparation, or PCR testing. An extensive search into potential contamination was performed for the 2019 juvenile raccoon case, since the most likely source of contamination can be positive samples processed around the same time. Rabies virus sequences from the 2019 juvenile raccoon displayed > 1% differences to sequences from all positive samples processed at PABOL within two weeks of the juvenile raccoon, suggesting cross contamination did not occur. 

Many laboratories in the United States already employ RT-qPCR as a confirmatory test for rabies when DFA exhibits non-specific staining. In these cases, RT-qPCR can confirm a negative result, avoiding unnecessary post-exposure prophylaxis for exposed humans or reducing the quarantine period for exposed animals. The findings from the two PA raccoon cases support expanding the role of RT-qPCR in rabies diagnosis in the United States. If RT-qPCR were routinely performed on all samples with DFA, it may improve sensitivity and increase the ability of laboratories to detect rabies cases with extremely sparse and non-uniform antigen distribution. However, not all rabies RT-PCR and RT-qPCR diagnostic tests have been thoroughly validated against the DFA to determine their diagnostic performance. OIE and WHO recommend pan-lyssavirus RT-PCR or RT-qPCR assays for rabies diagnostic testing [7,8].

Laboratory animals exhibit decreasing viral load from the brain stem to the forebrain during early non-mucosal rabies infections [23,24], which is very similar to what was observed in these two PA raccoon cases. It remains unclear if animals are capable of transmitting virus during very early infections. For rabies virus to be transmitted, it must travel from the central nervous system to the periphery, specifically to the nerve endings in the salivary glands. Once in the salivary glands, the rabies virus is secreted in saliva and can be transmitted by a bite. No rabies virus RNA was present in any of the seven samples collected from the salivary glands of the 2017 raccoon; tissue was not available for the 2019 juvenile raccoon.

The presence of rabies virus neutralizing antibodies can interfere with infection and lead to low viral levels in the brain, which could explain the low antigen and RNA levels observed. Animals can develop virus neutralizing antibodies after vaccination, and vaccinated animals with sub-protective immunity can succumb to rabies virus infection [25,26,27,28,29,30,31,32]. Oral rabies vaccination baits are distributed in western Pennsylvania as part of USDA’s raccoon rabies control program. The 2019 juvenile raccoon case was from Venango County in western PA, adjacent to the oral vaccination zone. It is possible that the 2019 juvenile raccoon was partially immunized, but not fully protected from rabies infection, possibly through inherited maternal antibodies. The 2017 adult raccoon was collected in Carbon County in eastern Pennsylvania; it is unlikely this raccoon encountered oral vaccine. However, even in rabies enzootic and epizootic areas without wildlife vaccination, wild animals have been shown to have neutralizing antibodies attributed to acquired immunity from sublethal exposures [31,33,34,35,36,37,38]. Unfortunately, serum samples were not available from either animal for testing.

In this study, the brain stem and rostral spinal cord were the most reliable tissues for rabies detection. Both DFA and PCR tests on cerebellum and hippocampus produced negative or indeterminate results for at least some replicates. The brain stem is one of the first brain structures where the rabies virus is observed in natural infections or after experimental inoculation in peripheral muscle or foot [39,40,41,42,43,44,45]. The increased reliability of brain stem and cerebellum for rabies diagnosis has been well-documented in the literature, and insufficient sampling can lead to false negative results [39,42,46,47,48,49,50]. For DFA, typical rabies antigen in the hippocampus and cerebellum can be more obvious due to large inclusions sometimes observed in pyramidal and Purkinje neuron somas [44]. In early infections, antigen may present as dust-like particles in the axon bundles of the brain stem; although, more frequently inclusions of all sizes are also present. DFA testing personnel should be familiar with both presentation types. A full cross section of the brain stem and tissue from the cerebellum or hippocampus is currently recommended for rabies diagnostic testing by WHO, OIE, and the US minimal national standard protocol [7,8,9,12]; the spinal cord is not recommended. It should be emphasized that neither DFA nor PCR can rule-out rabies if the required brain areas are not available or recognizable. However, due to the increased sensitivity, objectivity, and technical convenience of the LN34 RT-qPCR, it has potential to replace the DFA or serve as an alternative test for animal rabies diagnostics. Therefore, the PCR deserves further evaluation as the primary rabies test in the United States.

## Figures and Tables

**Figure 1 viruses-14-01845-f001:**
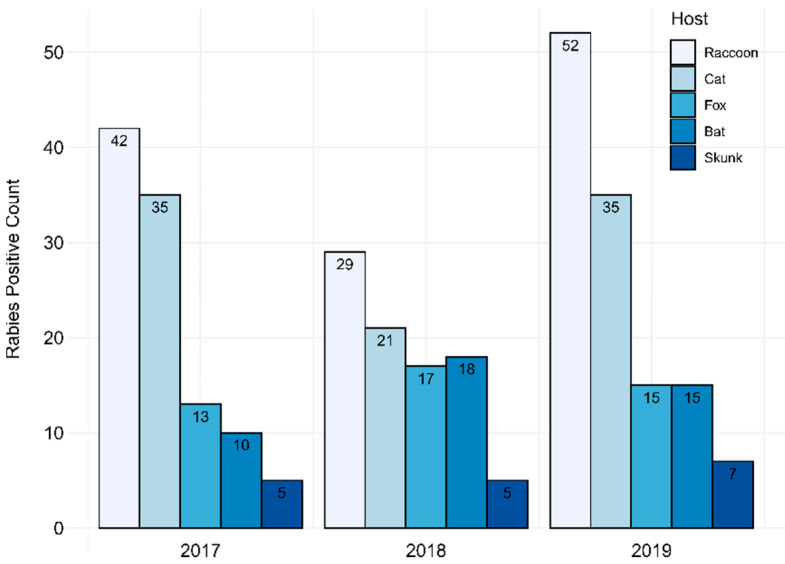
Distribution of positive rabies samples tested at PABOL during 2017–2019 by host animal. Raccoons accounted for 37% (42/113), 31% (29/95) and 39% (52/134) of positive cases each year, respectively.

**Figure 2 viruses-14-01845-f002:**
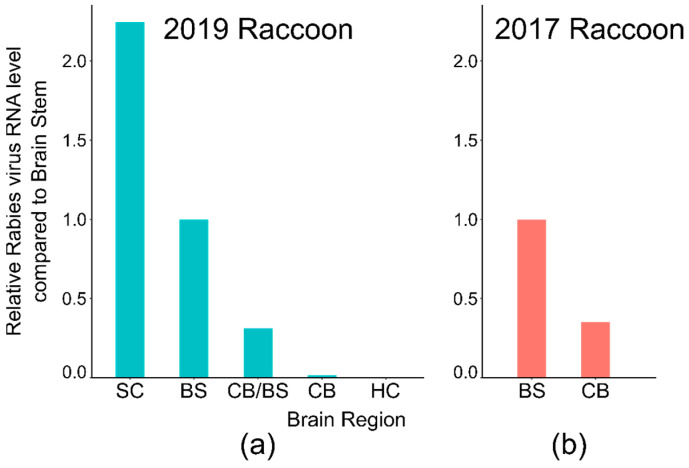
Rabies virus RNA level in different brain structures of PA raccoon cases. Relative rabies virus RNA level in different brain regions of 2019 (**a**) and 2017 (**b**) PA raccoon samples. Rabies virus RNA (LN34 Cq value) was normalized to beta actin level and compared to brain stem using the ΔΔCq method [13]. SC—spinal cord, BS—brain stem, CB/BS—mix of brain stem and cerebellum, CB—cerebellum, HC—hippocampus/cortex.

**Figure 3 viruses-14-01845-f003:**
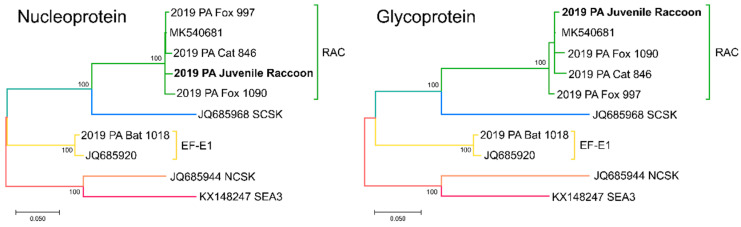
Phylogenetic trees showing clustering of PA 2019 juvenile raccoon rabies virus nucleoprotein (**left**) and glycoprotein (**right**) sequences with other rabies positive PA and reference sequences. Reference sequences from Eastern Raccoon (RAC), South Central Skunk (SCSK), *Eptesicus fuscus* Eastern 1 (EF-E1), North Central Skunk (NCSK) and Southeast Asia 3 (SEA3) rabies virus variants are shown with accession numbers. Branch color indicates variant: green is RAC, blue is SCSK, yellow is EF-E1, orange is NCSK and red is SEA3. The percentage of trees in which the associated taxa clustered together is shown next to the branches (based on 1000 bootstraps). Scale bar indicates number of substitutions per site.

**Table 1 viruses-14-01845-t001:** Rabies primers used for sequencing in this study.

Primer	Sequence
Nucleoprotein Forward	**TTTCTGTTGGTGCTGATATTGC**ACGCTTAACAACCAGATCAAAGAA TTTCTGTTGGTGCTGATATTGCACGCTTAACAACAAAATCADAGAAG
Nucleoprotein Reverse	**ACTTGCCTGTCGCTCTATCTTC**AGGAGGRGTGTTAGTTTTTTTC
Glycoprotein Forward	**TTTCTGTTGGTGCTGATATTGC**GATGTGAAAAAACTATYAACATCCCTC
Glycoprotein Reverse	**ACTTGCCTGTCGCTCTATCTTC**TGTGAKCTATTGCTTRTGTYCTTCA

Note: primers include 5′ sequence for adding Oxford Nanopore barcode sequences by PCR (bold).

**Table 2 viruses-14-01845-t002:** DFA and RT-qPCR results from PABOL.

		DFA Results			PCR Results		
		Result	Replicates	Antigen	Result	Replicates	Cq Value
2019 Juvenile Raccoon	Minced tissue *	Negative	0/1	ND	Positive	2/2	34.1
	Hippocampus	Negative	0/1	ND	NA	NA	NA
	Cerebellum	Negative	0/1	ND	Negative	0/2	ND
	Brain stem	Negative	0/1	ND	Indeterminate	2/2	35.5
	Spinal cord	Negative	0/1	ND	Positive	2/2	32.7
2017 Adult Raccoon	Minced tissue *	Negative	0/1	ND	Positive	8/8	32.1
	Cerebellum	Negative	0/1	ND	NA	NA	NA
	Brain stem	Negative	0/1	ND	NA	NA	NA
	Spinal cord	Negative	0/1	ND	Positive	2/2	34.2

Average rabies virus (LN34) Cq value is given for each sample, where lower Cq value indicates higher rabies virus RNA level. Cq value > 35 was used to define indeterminate result for LN34 RT-qPCR, based on previous publication [11]. NA: sample not available. ND: not detected. * Initial minced tissue from hippocampus, cerebellum, brain stem and spinal cord. Number of replicates where any antigen or RNA was detected are given over total replicates tested.

**Table 3 viruses-14-01845-t003:** DFA and RT-qPCR results from CDC.

		DFA Results			PCR Results		
		Result	Replicates	Antigen	Result	Replicates	Cq Value
2019 Juvenile Raccoon	Cortex/HC	Indeterminate	1/6	Atypical	Indeterminate	2/3	42.6 *
	Cerebellum	Positive	3/5	<10%	Indeterminate **	4/6	39.0 **
	Brain stem	Positive	3/3	<10%	Positive	3/3	35.0
	Spinal cord	NA	NA	NA	Positive	3/3	32.4
2017 Adult Raccoon	Cerebellum	Positive	2/6	<10%	Positive	3/3	31.8
	Brain stem	Positive	1/1	<10%	Positive	3/3	31.4
	Spinal cord	Positive	1/1	<10%	NA	NA	NA

Antigen distribution refers to percent of fields showing positive rabies antigen. Average rabies virus (LN34) Cq value is given for each sample, where lower Cq value indicates higher rabies virus RNA level. Cq value > 35 was used to define indeterminate result for LN34 RT-qPCR, based on previous publication [11]. NA sample not tested. * Average Cq values do not include replicates that did not produce Cq values (1/3 for cortex/hippocampus (HC) and 2/6 for cerebellum). ** For the cerebellum of the 2019 juvenile raccoon, only 1 out of 6 replicates produced a positive Cq value of <35, 3/6 produced Ct > 35 and 2/6 produced amplification that did not cross the threshold; the average Cq value for all the replicates was 39.0 making the final result indeterminate. Number of replicates where any antigen or RNA was detected are given over total replicates tested.

## Data Availability

Sequences have been deposited to GenBank under accession numbers OP221406-OP221415.

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
