# Peer review of "Detection of Apparent Early Rabies Infection by LN34 Pan-Lyssavirus Real-Time RT-PCR Assay in Pennsylvania"

_viruses, 2022, doi:10.3390/v14091845_

Round 1
Reviewer 1 Report
The article presents interesting observations based on the results of the diagnostic laboratory for three years. The results will be of interest for upgrading the protocols diagnosis of rabies virus. The data is presented completely and mostly correctly.
The article has a several of comments.
1. Figure 2 is questionable. PCR curves without normalization cannot serve as a basis for conclusions. Therefore Figure 2A can be omitted.
Figure 2B is not very readable and it is better to present it not as a curve, but as a histogram.
2. 207-211 lines: qPCR was mentioned for raccoon 2019 homogenate but result is missing.
3. I would like to see a more clearly expressed conclution with the main ideas and message to other researchers.
Author Response
We would like to thank the reviewer for the time and effort in reviewing our manuscript.
Reviewer 1
The article presents interesting observations based on the results of the diagnostic laboratory for three years. The results will be of interest for upgrading the protocols diagnosis of rabies virus. The data is presented completely and mostly correctly.
The article has a several of comments.
- Figure 2 is questionable. PCR curves without normalization cannot serve as a basis for conclusions. Therefore Figure 2A can be omitted.
Figure 2B is not very readable and it is better to present it not as a curve, but as a histogram.
Thank you. Figure 2A has been removed and Figure 2B was converted to a bar graph, as suggested.
- 207-211 lines: qPCR was mentioned for raccoon 2019 homogenate but result is missing.
We have removed that statement from the text.
- I would like to see a more clearly expressed conclusion with the main ideas and message to other researchers.
The following statements were added to address the reviewer’s request: “However, due to the increased sensitivity, objectivity, and technical convenience of the LN34 RT-qPCR, it has potential to replace the DFA or serve as an alternative test for animal rabies diagnostics. Therefore, the PCR deserves further evaluation as the primary rabies test in the United States.”
Reviewer 2 Report
If it is possible you should make biological test in mice for checking your results, just in the two racoon cases.
Author Response
We would like to thank the reviewer for the time and effort in reviewing our manuscript.
Reviewer 2
If it is possible you should make biological test in mice for checking your results, just in the two racoon cases.
We thank the reviewer for this suggestion; however, this experiment is outside the scope of the current study.
Round 2
Reviewer 2 Report
My resolution is to accept this MS in the present way.